# Transfer Learning for Related Reinforcement Learning Tasks via Image-to-Image Translation

## Abstract

Deep Reinforcement Learning has managed to achieve state-of-the-art results in learning control policies directly from raw pixels. However, despite its remarkable success, it fails to generalize, a fundamental component required in a stable Artificial Intelligence system. Using the Atari game Breakout, we demonstrate the difficulty of a trained agent in adjusting to simple modifications in the raw image, ones that a human could adapt to trivially. In transfer learning, the goal is to use the knowledge gained from the source task to make the training of the target task faster and better. We show that using various forms of fine-tuning, a common method for transfer learning, is not effective for adapting to such small visual changes. In fact, it is often easier to re-train the agent from scratch than to fine-tune a trained agent. We suggest that in some cases transfer learning can be improved by adding a dedicated component whose goal is to learn to visually map between the known domain and the new one. Concretely, we use Unaligned Generative Adversarial Networks (GANs) to create a mapping function to translate images in the target task to corresponding images in the source task. These mapping functions allow us to transform between various variations of the Breakout game, as well as between different levels of a Nintendo game, Road Fighter. We show that learning this mapping is substantially more efficient than re-training. A visualization of a trained agent playing Breakout and Road Fighter, with and without the GAN transfer, can be seen in `https://streamable.com/msgtm` and `https://streamable.com/5e2ka`.

## 1 Introduction

Transferring knowledge from previous occurrences to new circumstances is a fundamental human capability and is a major challenge for deep learning applications. A plausible requirement for artificial general intelligence is that a network trained on one task can reuse existing knowledge instead of learning from scratch for another task. For instance, consider the task of navigation during different hours of the day. A human that knows how to get from one point to another on daylight will quickly adjust itself to do the same task during night time, while for a machine learning system making a decision based on an input image it might be a harder task. That is because it is easier for us to make analogies between similar situations, especially in the things we see, as opposed to a robot that does not have this ability and its knowledge is based mainly on what it already saw.

Deep reinforcement learning has caught the attention of researchers in the past years for its remarkable success in achieving human-level performance in a wide variety of tasks. One of the field's famous achievements was on the Atari 2600 games where an agent was trained to play video games directly from the screen pixels and information received from the game (Mnih et al., 2013). However, this approach depends on interacting with the environment a substantial number of times during training. Moreover, it struggles to generalize beyond its experience, the training process of a new task has to be performed from scratch even for a related one. Recent works have tried to overcome this inefficiency with different approaches such as, learning universal policies that can generalize between related tasks (Schaul et al., 2015), as well as other transfer approaches (Fernando et al., 2017; Rusu et al., 2016).

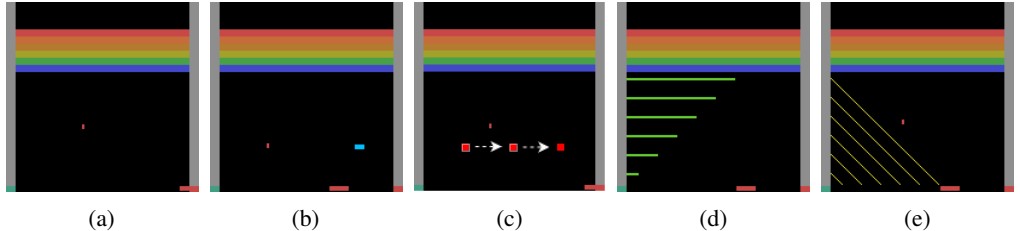

(a)  (b)  (c)  (d)  (e)

Figure 1: Various variations of the Breakout game: (a) Standard version, (b) A Constant Rectangle - a rectangle in the same size as the bricks is added to the background in a predefined location, (c) A Moving Square - a square is added to the background and its location changes to one of three predefined locations every 1000 steps, (d) Green Lines - green lines in different sizes are drawn in the background, (e) Diagonals - diagonals are drawn in the left side of the background.

In this work, we first focus on the Atari game Breakout, in which the main concept is moving the paddle towards the ball in order to maximize the score of the game. We modify the game by introducing visual changes such as adding a rectangle in the middle of the image or diagonals in the background. From a human perspective, it appears that making visual changes that are not significant to the game's dynamics should not influence the score of the game, a player who mastered the original game should be able to trivially adapt to such visual variants. We show that the agent fails to transfer. Furthermore, fine-tuning, the main transfer learning method used today in neural networks, also fails to adapt to the small visual change: the information learned in the source task does not benefit the learning process of the very related target task, and can even decelerate it. The algorithm behaves as if these are entirely new tasks.

Our second focus is attempting to transfer agent behavior across different levels of a video game: can an agent trained on the first level of a game use this knowledge and perform adequately on subsequent levels? We explore the Nintendo game Road Fighter, a car racing game where the goal is to finish the track before the time runs out without crashing. The levels all share the same dynamics, but differ from each other visually and in aspects such as road width. Similar to the Breakout results, an agent trained to play the first level fails to correctly adapt its past experience, causing the learned policy to completely fail on the new levels.

To address the generalization problem, we propose a zero-shot generalization approach, in which the agent learns to transfer between related tasks by learning to visually map images from the target task back to familiar corresponding images from the source task. Such mapping is naturally achieved using Generative Adversarial Networks (GANs) (Goodfellow et al., 2014), one of the most popular methods for the image-to-image translation that is being used in computer vision tasks such as style transfer (Zhu et al., 2017; Kim et al., 2017), object transfiguration (Zhou et al., 2017), photo enhancement (Ledig et al., 2016) and more recently, video game level generation (Volz et al., 2018). In our setup, it is not realistic to assume paired images in both domains, calling for the use of Unaligned GANs (Liu et al., 2017; Zhu et al., 2017; Kim et al., 2017; Yi et al., 2017). Using this approach we manage to transfer between similar tasks with no additional learning.

**Contributions** This work presents three main contributions. First, in Section 2, we demonstrate how an agent trained with deep reinforcement learning algorithms fails to adapt to small visual changes, and that the common transfer method of fine-tuning fails as well. Second, in Section 3, we propose to separate the visual mapping from the game dynamics, resulting in a new transfer learning approach for related tasks based on visual input mapping. We evaluate this approach on Breakout and Road Fighter, and present the results comparing to different baselines. We show that our visual transfer approach is much more sample efficient then the alternatives. Third, in section 5, we suggest an evaluation setup for unaligned GAN architectures, based on their achieved performance on concrete down-stream tasks.

## 2 GENERALIZATION FAILURES OF DEEP RL

Many Breakout variations can be constructed that involve the same dynamics. The main idea is to make modifications that are not critical for a human playing the game but are for the algorithm that

relies on visual inputs. We demonstrate the difficulty of deep reinforcement learning to generalize using 4 types of modifications as presented in Figure 1.

## 2.1 SETUP

For all the experiments in this section forward we use the Asynchronous Advantage Actor-Critic (A3C) algorithm (Mnih et al., 2016), taking advantage of being faster than Deep Q-Network (DQN) (Mnih et al., 2013). The A3C learns the policy and the state-value function using parallel actor-learners exploring different policies for the acceleration and stability of the training.

We rescale the image to $80 \times 80$ and keep the RGB colors for more realistic images. We use 32 actor learners, a discount rate of 0.99, learning rate of 0.0001, 20-step returns, and entropy regularization weight of 0.01. The A3C variation we choose is the LSTM-A3C network. We use the standard high-performance architecture implemented in (Kostrikov, 2018).

## 2.2 TRANSFER-LEARNING VIA FINE-TUNING

The setup mentioned in 2.1 successfully trains on Breakout, reaching a score of over 400 points. However, when a network trained on the original game is presented with the game variants, it fails completely, reaching to a maximum score of only 3 points. This shows that the network does not necessarily learn the game's concepts and heavily relies on the images it receives.

The common approach for transferring knowledge across tasks is fine-tuning. We experiment with common techniques used in deep learning models. In each setting, we have a combination of frozen and fine-tuned layers (Partial/Full) as well as layers that are initialized with the target's parameters and layers that are initialized with random values (Random). Our settings are inspired by (Yosinski et al., 2014). We train each one of the tasks (before and after the transformation) for 60 million frames, and our evaluation metric is the total reward the agents collect in an episode averaged by the number of episodes, where an episode ends when the game is terminated or when a number of maximum steps is reached. We periodically compute the average during training. We consider the following settings:

- From-Scratch: The game is being trained from scratch on the target game.
- Full-FT: All of the layers are initialized with the weights of the source task and are fine-tuned on the target task.
- Random-Output: The convolutional layers and the LSTM layer are initialized with the weights of the source task and are fine-tuned on the target task. The output layers are initialized randomly.
- Partial-FT: All of the layers are initialized with the weights of the source task. The three first convolutional layers are kept frozen, and the rest are fine-tuned on the target task.
- Partial-Random-FT: The three first convolutional layers are initialized with the weights of the source task and are kept frozen, and the rest are initialized randomly.

## 2.3 RESULTS

The results presented in Figure 2 show a complete failure of all the fine-tuning approaches to transfer to the target tasks. In the best scenarios the transfer takes just as many epochs as training from scratch, while in other cases starting from a trained network makes it *harder* for the network to learn the target task. As the graphs show, some of the modification influenced more than others. For example, Figure 2a shows that adding a simple rectangle can be destructive for a trained agent: while training from scratch consistently and reliably achieves scores over 300, the settings starting from a trained agent struggle to pass the 200 points mark within the same number of iterations, and have a very high variance. We noticed that during training the agent learns a strategy to maximize the score with a minimum number of actions. None of the experiments we performed showed better results when the layers in the network were fine-tuned, and some showed negative transfer which is a clear indication of an overfitting problem. The A3C model learned the detail and noise in the training data to the extent that it negatively impacted the performance of the model on new data. Our results and conclusions drawn from them are consistent with the results shown when a similar approach was

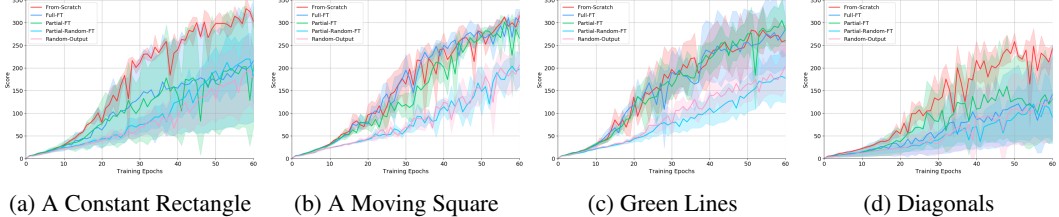

| (a) A Constant Rectangle | (b) A Moving Square | (c) Green Lines | (d) Diagonals |

Figure 2: A comparison between the different baselines on Breakout. The y-axis on each one of the plots shows the average reward per episode of Breakout during training. The x-axis shows the total number of training epochs where an epoch corresponds to 1 million frames. The plots are averaged on 3 runs with different random seeds. Each curve is the average and its background is the standard deviation.

used on Pong (Rusu et al., 2016). In addition to Breakout, we also experimented transfer between the first and advanced level of Road Fighter, where the backgrounds change but the dynamics remains the same. This experiments resulted with 0 points on each of the levels, a complete failure of the agent to re-use the driving techniques learned on the first levels on the next ones.

# 3 ANALOGY-BASED ZERO-SHOT GENERALIZATION

An agent capable of performing a task in a source domain is now presented with a new domain. Fine-tuning the agent on the target domain fails to transfer knowledge from the source domain. We propose to separate the visual transfer from the dynamics transfer. To perform well, the agent can try and make analogies from the new domain to the old one: after observing a set of states (images) in the new domain, the agent can learn to map them to similar, familiar states from the source domain, and act according to its source domain policy on the mapped state.

More concretely, given a trained policy $\pi(a|s; \theta)$ with trained parameters $\theta$ proposing an action $a$ for source domain states $s \in \mathcal{S}$, we wish to learn a mapping function $G : \mathcal{T} \mapsto \mathcal{S}$ from target domain states $t \in \mathcal{T}$ such that interacting with the environment $\mathcal{T}$ by applying the policy $\pi(a|G(t); \theta)$ will result in a good distribution of actions for the states $\mathcal{T}$, as indicated by high overall scores. In other words, we seek a mapping function $G$ that allows us to re-use the same policy $\pi_\theta$ learned for source environment $\mathcal{S}$ when interacting with the target environment $\mathcal{T}$.

As both the source and target domain items are images, we heuristically learn the function $G$ by collecting sets of images from $\mathcal{S}$ and $\mathcal{T}$ and learning to visually map between them using Unaligned GAN (Liu et al., 2017; Zhu et al., 2017; Kim et al., 2017; Yi et al., 2017). We use the scores obtained from interacting with the environment via $\pi(a|G(t); \theta)$ for the GAN model selection and stopping criteria.

## 3.1 UNSUPERVISED IMAGE-TO-IMAGE TRANSLATION

In this work, we focus on learning setups that receive only raw image data, without additional domain knowledge about objects or game dynamics. This prohibits us from using supervised paired GANs Isola et al. (2016) for learning the mapping function $G$: we cannot collect the needed supervision of corresponding $(s, t)$ pairs. Instead, we use unaligned GANs (Zhu et al., 2017; Liu et al., 2017; Kim et al., 2017; Yi et al., 2017), in which the learner observes two sets of images, one from each domain, with the goal of learning to translate images in one domain to images in another.

All major approaches to the unaligned image-to-image translation use the Cycle-Consistency principle. We have two mapping (encoding) functions $G_1 : T \mapsto S$ and $G_2 : S \mapsto T$ where $S = \{s_i\}_{i=1}^{N}$ is a set of images collected from the source task and $T = \{t_j\}_{j=1}^{M}$ is a set of images collected from the target task. The goal is to generate an image $s'$, for any given $t \in T$ where $G_1(t) = s'$, that is indistinguishable from $s \in S$. The cycle consistency principle relies on the assumption that the two functions, $G_1$ and $G_2$ are inverses of each other. It encourages unsupervised mapping by forcing $G_2(G_1(t)) = t$ and $G_1(G_2(s)) = s$ where $s$ and $t$ are the input images. The second component

of the GAN architecture are the discriminators $D_1$ and $D_2$ aiming to distinguish between images generated by $G_1$ and $G_2$ and the real images from the target and source distributions respectively.

In the following experiments, we use the UNIT framework (Liu et al., 2017), which we found to perform well for the Breakout tasks (in section 5 we explicitly compare the UNIT and CycleGAN approaches on both the Breakout and Road Fighter transfer tasks). A distinguishing element in the UNIT framework is the shared-latent space assumption, according to which there is a shared-latent space consisting a shared latent code $z$ for any pair of images $s$ and $t$ that can be recovered from this code. This share-latent space is represented as the weights of the last few layers of the encoding network and the few first layers of the decoding networks, and is learned by using Variational Autoencoders (VAEs). This sharing strongly ties the images in the source and target domain to each other, encouraging mappings that preserve similarities across domains. In contrast, the CycleGAN architecture (Zhu et al., 2017) does not make the shared space assumption and instead the generators are trained independently with two separate networks. For further information of the unaligned GAN architectures, see the original papers.

## 3.2 GAN Training

**Datasets.** The Unaligned GAN training dataset requires images from both domains. We collect images from the source domain by running an untrained agent and collecting the observed images, and we do similarly for the target domain. The number of collected images should balance between two objectives: On the one hand, we want to take a small number of images, and on the other hand, it is essential for us to have a diverse dataset. We repeat this procedure for every target task, and create a source-target dataset for each. During training, we further ensure the images pairs are not aligned by randomly picking an image from each set at each iteration.

**Setup and Analysis.** For our experiments we use the same architecture and hyper-parameters proposed in the UNIT paper. We initialize the weights with Xavier initialization (Glorot & Bengio, 2010), set the batch size to 1 and train the network for a different number of iterations on each task. Some tasks are harder than others, the more changes exist in the frames the harder it is for the GAN to learn the mapping between the domains. However, our evaluation metric, testing the agent with the generated images, is a clear indication of how hard each task is and the number of iterations needed is based on the results of this evaluation.

**Evaluation.** We use GAN training to learn a mapping function $G$. GAN training, and unaligned GANs training in particular, are unstable and it challenging to find a good loss-based stopping criteria for them. A major issue with GANs is the lack of an evaluation metric that works well for all models and architectures, and which can assist in model selection. Different works use different methods that were suitable for their types of data. Our setup suggests a natural evaluation criteria: we run the source agent without any further training while using the model to translate each image of the target task back to the source task and collect the rewards the agent receives during the game when presented with the translated image. We use the total accumulated rewards (the score) the agent collects during the game as our model accuracy.

## 4 Experiments

We examine how well the agent does when receiving translated frames generated by the generator trained with GANs. Since we initialized the layers with the values of the trained network, we assume that the success of the agent is dependent on the similarity between the generated and the source task's frames. We first test our approach on Breakout, evaluating its ability to remove the changes added in the images. Second, we challenge our method even more on Road Fighter, where the goal is to transfer between different environments.

### 4.1 Breakout

Our goal in Breakout is removing the modifications in each one of the target tasks and transfer between each to the original game. Although the variations share many similarities, some tasks were more challenging than others, e.g., the lines of the Green Lines variation hide parts of the ball in

Table 1: The score and number of frames needed for it of: the source task (Source), target task when initialized with the source task' network parameters with no additional training (Target) and the target task when initialized with the source task' network parameters where every frame is translated to a frame from the source task (Target with GANs).

| Source | | Target Task | Target | | Target with GANs | |
|---|---|---|---|---|---|---|
| Frames | Score | | Frames | Score | GAN iterations | Score |
| 43M | 302 | A Constant Rectangle | 0 | 3 | **260K** | **362** |
| 43M | 302 | A Moving Square | 0 | 0 | **384K** | **300** |
| 43M | 302 | Green Lines | 0 | 2 | **288K** | **300** |
| 43M | 302 | Diagonals | 0 | 0 | **380K** | **338** |

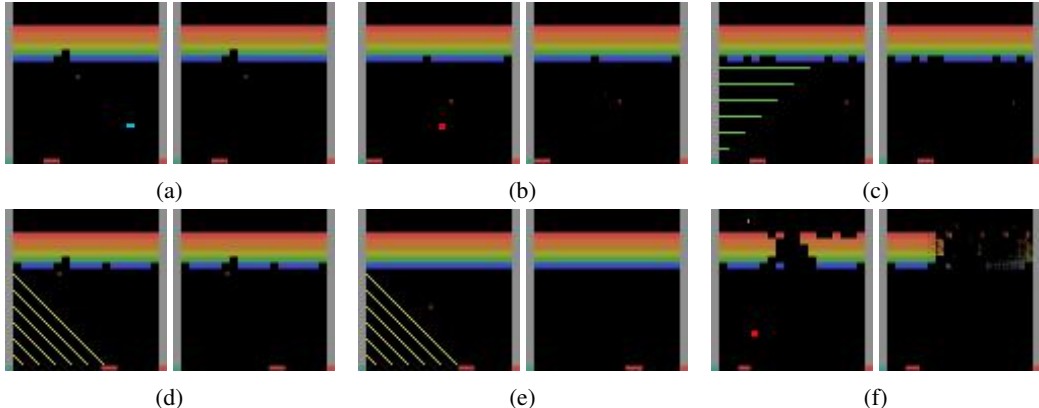

(a)      (b)      (c)

(d)      (e)      (f)

Figure 3: Illustration of a frame taken from the target task (left) and its matching frame of the source task generated with GANs (right) for each one of the Breakout variations. (a)-(d) demonstrate successes, while (e) and (f) show failure modes of the unaligned GAN. In (e) the ball in the input image is not generated in the output and in (f) not all bricks are generated, and some of the generated bricks appear smudged.

some frames. On the opposite side, the Rectangle variation requires less training since the number of pixels changed in the image is small.

During testing, we encountered problems with the images generation that we did not observe during the GAN training. The translation task we attempted to perform was supposedly simple – search for the differences between the domains shared by all images, change them to the way they are in the opposite domain and leave everything else the same. Unfortunately, since the network does not have any prior information about objects in the image, it struggles to generate them even if they were not changed. The most common problem we had was that the generator generated bricks that were supposed to be removed from the game, and in some cases, they were very noisy in the generated image (Fig. 3f). Another problem was the ball generation and more specifically, the location of the generated ball. Since the ball is small and changes its position often, it was hard for the generator, trained with unaligned pairs, to decide if and where to locate it in the generated image (Fig. 3e). These issues and others eventually caused the agent to fail in following the policies it learned on the source task. We found that more training leads to better results for some of the variations and so the number of iterations needed was different for each variation.

In Table 1 we show the results of a test game played by the agent with and without GANs. We stop the training after reaching 300 points, which we consider to be a high score. As the results show, the source game trained from scratch requires ten of millions of images to achieve such score comparing to the target task trained with GANs that only needs a few hundreds of thousands—a 100x fold increase in sample efficiency. Moreover, the frames the GAN was trained on were limited to the first games in which the A3C network was not trained, and yet it managed to generalize to more advanced stages of the game.

## 4.2 ROAD FIGHTER

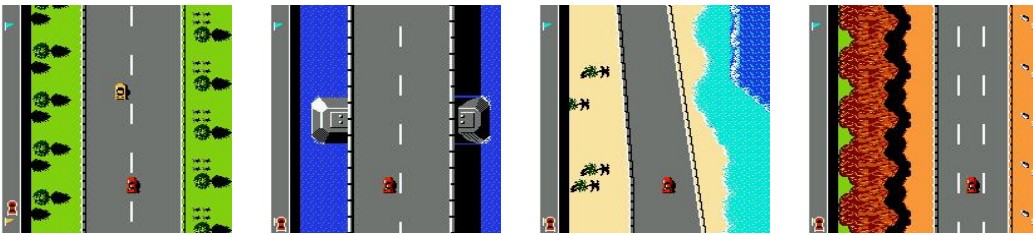

Figure 4: Road Fighter levels from left to right: Level 1, Level 2, Level 3 and Level 4.

While the Breakout variants work well to demonstrate transfer failure cases, they can be considered as "toy examples". We proceed to demonstrate the effectiveness of our transfer method on a "real" task: learning to transfer between different levels of the Road Fighter game. Road Fighter contains 4 different levels (Fig. 4), each with a different background where some are more difficult than others. The levels mostly differ visually and all have the same rules and require the same driving techniques. Thus, we believe that these techniques should sustain when playing a new level. We start by training an RL agent to play the first level of the game. To maximize the score, the agent has to acquire 3 main capabilities: driving fast, avoiding collision with obstacles, and if a car collision occurs reacting fast to avoid crashing. We use the A2C algorithm, the synchronous version of the Advantage Actor-Critic which performs better on this game than A3C, reaching over $10,000$ game points on level 1.

For the GAN training, we collect $100k$ $84x84$ frames from each of levels 2, 3 and 4 by running an untrained agent repeatedly until we have collected sufficient samples (training an RL agent to reach a score of 10,000 points on Level 1 required observing above 100M frames.). Using the collected images we train a mapping function for each task to map the new level (target task) to the first one (source task). We use the same GAN architecture used for Breakout, but initialize the weights with Orthogonal initialization. Compared to Breakout, these tasks introduce new challenges: rather than removing a mostly static element, the GAN has to be able to change the background and road size while keeping the cars in the right position on the road. On the other hand, this setup may be closer to the one unaligned GANs are usually applied in. We restrict ourselves to collecting images from the beginning of the game, before the agent had any training. This restricts the phenomena the GAN can observe, and some target tasks' images do not have a clear corresponding situation in the first level, potentially causing unpredictable behaviors. For example, the generator matches the diagonal shaped roads to one of the first rare and unique images of level 1 (Fig. 5e).

Our experiments presented in Table 2 demonstrate how an agent trained to master the first level of the game fails to follow the optimal policies on new levels, reaching 0 points. However, with the GAN-based visual analogies the agent is able to apply some of the abilities it gained when training on the first level, most notably driving fast, staying on the road, avoiding some cars, and, most importantly, recovering from car crashes. The resulting agent achieves impressive scores on levels 2, 3 and 4 (5350, 5350 and 2050 points, respectively), with no additional RL training and while observing only a fraction of the frames required for training a corresponding RL agent from scratch for these levels.

**Limitations.** While the GAN works well in generating objects it has seen before, such as the agent's car, it does have limitations on objects it has not seen. As a result, it ends up generating differently colored cars all in red, or not generating them at all, as shown in Fig. 5a, 5d and 5f. Colorful cars can be "collected" by the agent and are worth 1000 points each. Generating them in red makes the agent avoid them, losing these extra points and achieving overall lower scores even if finishing the track. When cars are not fully generated, the agent is less likely to avoid them, and eventually crashes.

**Data Efficiency.** We measure the number of frames of game-interaction needed for the analogy-transfer method. We collect 100k frames, and then train the GAN for up to $500k$ iterations, evaluating it every $10,000$ iterations by running the game and observing the score, and pick the best scoring model. This amounts to $100k + 50 * F$ frames, where $F = 3000$ is roughly the average number of frames in a game. This amounts to about $250k$ frames of game interaction for each transferred level,

Table 2: The scores of the agent on every level of Road Fighter with and without analogy-transfer, as well as the number of game-interaction frames needed for for the analogy transfer and for achieving a similar score with RL training.

|  | Score | Score with Analogy Transfer | # Frames | # Frames needed with Full Training |
| --- | --- | --- | --- | --- |
| Level 2 | 0 | 5350 | 250K | 12.4M |
| Level 3 | 0 | 5350 | 250K | >50M |
| Level 4 | 0 | 2050 | 250K | 13.6M |

Figure 5: Left: the original frame. Right: GAN generated. Upper row shows the success cases of the GAN while the lower row shows representative failures: in (d) and (f) the only object generated on the road is the player's car and in (e) the diagonal shaped road of level 2 in matched to the starting point of level 1.

an order of magnitude fewer interaction frames than training an RL agent to achieve a comparable score.

**Discussion.** The approach is successfully transferring to most tasks, achieving the best scores on levels 2 and 3. In general, as the game progresses, the more obstacles are presented making it harder to play. One of our best results is achieved on level 2 where the road is identical to level 1's, reaching the best score after $320k$ GAN iterations. On level 3, the challenge increases as the road is very narrow, making it harder for a player to avoid crashing. However, the GAN manages to generate the road in the right shape in most frames and position the cars in the matching ratio. Moreover, despite the challenges, due to the agent's ability to avoid crashing when colliding with cars it gets over $5000$ points after $450k$ GAN iterations. In level 4 we get the maximum after $270k$ iterations. This level is also the most challenging one to play, which might be the reason for the score being the lowest out of all tasks. The main obstacles are the sudden turns in the road causing the car to be very close to the sideways and the increasing dangers a player has to avoid. Theses difficulties make this level much harder than level 1 and might require more training even from a human player.

Overall, the agent performs quite well by successfully applying 2 out of 3 main capabilities it gained during training. It is missing the third capability, avoiding collisions and collecting bonus cars, mainly because of bad generation. We believe that these tasks and results demonstrate a success of the analogy transfer method for zero-short generalization across different levels of a video game. They also suggest a potential of performing well on additional real world tasks in which visual analogies can be made.

## 5 TOWARDS TASK-ORIENTED GAN EVALUATION

Evaluating GAN models and comparing them to each other is a major challenge: working with images and without well-defined objectives, testing the quality of the generator is delegated to human judgment, often using crowdsourcing to evaluate the generated images Isola et al. (2016); Donahue et al. (2017). This metric is not stable and can be unreliable due to changing human factors. Others

use a linear classifier to evaluate the image representations the GAN learned on supervised datasets, e.g., MNIST and CIFAR-10 (Radford et al., 2015). These approaches and others may be good enough for a limited group of test images but do not necessarily reflect the performance of a model in the real world.

We note that while unaligned GANs are known to achieve impressive results with only few training examples, our seemingly trivial translation cases proved to be quite challenging for the unaligned GAN architectures: producing valid translations that work in the game require training on a substantial number of images. During training, the GAN sees images from the early stages of the game where only a few bricks are missing in Breakout and there are no obstacles in Road Fighter. Our task requires it to generalize by translating images where objects are removed, added or in a different locations. This requires levels of generalization that may not be reflected in existing test-sets. We argue that using simple, well defined yet diverse test cases, situated in the context of a concrete down-stream task like we do here, is an important step forward in the evaluation of GAN models.

We propose to evaluate GAN by running a game in which the trained Deep RL network that learned policies based on images is now receiving images from the same domain generated with GAN, and equate a successful GAN model with one resulting in a high game score. We use the approach to compare Cycle-GAN (Zhu et al., 2017) and UNIT-GAN (Liu et al., 2017).

We examine both methods by running the RL agent with each every 1000 GAN training iterations and considering the maximum score after $500k$ iterations. We present the results in Table 3. The UNIT GAN performs better than Cycle-GAN in most Breakout tasks, while Cycle-GAN outperforms UNIT in most Road Fighter tasks while requiring fewer iterations. The main difference between the two methods is the weight-sharing constraint applied in UNIT, making the domains dependent on each other by sharing and updating the weights of one or several decoders and encoders layers. We hypothesize this constraint is an advantage in tasks where the representation of the images in the different domains are similar. Thus, in Breakout, where most pixels are identical in the source and target images, Cycle-GAN often fails where UNIT succeed. However, in tasks such as Road Fighter's where most pixels are different, the agent could benefit from architectures such as Cycle-GAN where the two domains are independent of each other.

## 6  RELATED WORK

Transfer Learning (TL) is a machine learning technique used to improve the training speed of a target task with knowledge learned in a source task. Pretraining and fine-tuning was proposed in Hinton & Salakhutdinov (2006) and applied to TL in Bengio (2012) and Dauphin et al. (2012). In this procedure, the approach is to train the base network and then copy its first $n$ layers to the first $n$ layers of a target network. One can choose to update the feature layers transferred to the new task with the error backpropagated from its output, or they can be left frozen, meaning that they do not change during training on the new task. Unfortunately, as we have shown, while fine-tuning might have the ability to accelerate the training process is some cases, it can also have a damaging impact on others.

Generalization is a key element in training deep learning models with time or data size constraints. Recent discussions on overfitting in Deep RL algorithms (Zhang et al., 2018) encouraged better evaluation (e.g. OpenAI Retro Contest [1]) and generalization methods. In Atari, there are many similarities between the goals and the mechanism of the games. For this reason, there have been many works attempting to transfer between games or between different variations of the same game, one approach trying to do both is the *progressive networks* (Rusu et al., 2016). A progressive network is constructed by successively training copies of A3C on each task of interest. In that work, they transferred between different games as well as from different variations of the game Pong. The drawback of this approach is the number of parameters growing quadratically with the number of tasks. However, even if this growth rate was improved, different tasks may require different adjustments and the predefinition of the number of layers and network representation is preventing it.

Zero-shot generalization is a discussed and researched topic nowadays. One work transferring between modified versions of the same game using zero-shot transfer is the schema networks (Kansky et al., 2017). Like us, they also chose to demonstrate their method on the game Breakout, using

---

[1] https://contest.openai.com/

Table 3: The scores accumulated by an Actor-Critic RL agent using UNIT and Cycle-GAN.

| Method | UNIT | | CycleGAN | |
|---|---|---|---|---|
| | Frames | Score | Frames | Score |
| A Constant Rectangle | 333K | 399 | 358K | 26 |
| A Moving Square | 384K | 300 | 338K | 360 |
| Green Lines | 378K | 314 | 172K | 273 |
| Diagonals | 380K | 338 | 239K | 253 |
| Road Fighter - Level 2 | 274K | 5750 | 51K | 6000 |
| Road Fighter - Level 3 | 450K | 5350 | 20K | 3200 |
| Road Fighter - Level 4 | 176K | 2300 | 102K | 2700 |

Object Oriented Markov Decision Process. In contrast, we do not use the representation of the objects in the game, and we wish to preserve the accomplishments of DQN and transfer using only raw data. Other attempted to achieve robust policies using learned disentangled representation of the image (Higgins et al., 2017), analogies between sets of instructions (Oh et al., 2017), *interactive replay* (Bruce et al., 2017) while training and learn general policies by training on multiple tasks in parallel (Espeholt et al., 2018; Sohn et al., 2018).

Finally, the idea of using GANs for transfer learning and domain adaptation was explored for supervised image classification and robotics applications by several authors (Bousmalis et al., 2016; Hoffman et al., 2017; Liu & Tuzel, 2016; Bousmalis et al., 2017). In these methods, there is supervised source domain data $(x_i^s, y_i^s)$ and unlabeled target domain data, and the GAN variant $G$ is trained to map source samples $x_i^s$ to target-like samples $G(x_i^s)$ . Then, a classifier is trained on the generated data $(G(x_i^s), y_i^s)$. Our RL-based setup is different: first, our coverage of target-domain data is very limited (we can only observe states which are reachable by the un-adapted or untrained agent). Second, we do not have access to supervised gold labels on the source domain, but only to a learned policy network. Third, interactions with the game environment provide very indirect rewards, so using this reward signal to influence the GAN training will be very inefficient. We thus opt for a different strategy: rather than mapping the source to the target domain and training on the projected signal, which is unrealistic an costly in the RL setup, we instead take a pre-trained source model and train an unaligned GAN to map from the target domain back to the source domain, in order to re-use the source model's knowledge and apply it to the target domain data. We believe this form of usage of GAN for transfer learning is novel.

## 7 CONCLUSIONS AND FUTURE WORK

We demonstrated the lack of generalization by looking at artificially constructed visual variants of a game (Breakout), and different levels of a game (Road Fighter). We further show that transfer learning by fine-tuning fails. The policies learned using model-free RL algorithms on the original game are not directly transferred to the modified games even when the changes are irrelevant to the game's dynamics.

We present a new approach for transfer learning between related RL environments using GANs without the need for any additional training of the RL agent, and while requiring orders of magnitude less interactions with the environment. We further suggest this setup as a way to evaluate GAN architectures by observing their behavior on concrete tasks, revealing differences between the Cycle-GAN and UNIT-GAN architectures. We believe our approach is applicable to cases involving both direct and less direct mapping between environments, as long as an image-to-image translation exist. While we report a success in analogy transfer using Unaligned GANs, we also encountered limitations in the generation process that made it difficult for the agent to maximize the results on the Road Fighter's tasks. In future work, we plan to explore a tighter integration between the analogy transfer method and the RL training process, to facilitate better performance where dynamic adjustments are needed in addition to the visual mapping.

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
