# OpenReview forum: "Transfer Learning for Related Reinforcement Learning Tasks via Image-to-Image Translation"
_ICLR.cc/2019/Conference_

### Official Review · AnonReviewer1 · 2018-11-02
**Problematic qualitative results, generic unsupervised domain adaptation**

**Rating:** 4
**Confidence:** 4

**Review:**

# Summary

This paper proposes to improve the sample efficiency of transfer learning for Deep RL by mapping a new visual domain (target) onto the training one (source) using GANs. First, a deep RL policy is trained on a source domain (e.g., level 1 of the Atari Road Fighter game). Second, a GAN (e.g. UNIT or CycleGAN) is trained for unsupervised domain adaptation from target images (e.g., level 2 of Road Fighter) to source ones. Third, the policy learned in the source domain is applied directly on the GAN-translated target domain. The experimental evaluation uses two Atari games: i) transfer from Breakout to Breakout with static visual distractors inpainted on the screen, ii) from one Road Fighter level to others. Results suggest that this transfer learning approach requires less images than retraining from scratch in the new domain, including when fine-tuning does not work.


# Strengths

Controlled toy experiments of Deep RL generalization issues:
The experiments on Breakout quantify how badly A3C overfits in this case, as it shows catastrophic performance degradation even with trivial static visual input perturbations (which are not even adversarial attacks). The fine-tuning experiments also quantify well how brittle the initial policy is, motivating further the importance of the problem studied by the paper.

Investigating the impact of different GANs on the end task:
The experiments evaluate two different image translation algorithms: one based on UNIT, the other based on CycleGAN. The results suggest that this choice is key and depends on the target domain. This suggests that the adaptation is in fact task dependent, confirming the direction pursued by others in task-specific unsupervised domain adaptation (cf. below).


# Weaknesses

Discrepancy between quantitative and qualitative results:
The good quantitative results (accumulated rewards) reported in the experiments are not reflected in the qualitative results. As can be seen from the videos, these results seem more to be representative of a bias in the data. For instance, in the Road Fighter videos, one can clearly see that the geometry of the road (width, curves) and dynamic obstacles are almost completely erased in the image translation process. The main reasons the quantitative results are good seem to be i) in the non-translated case the agent crashes immediately, ii) the "translated" image is a wide straight road identical to level 1 where the policy just keeps the car in the middle (thus crashing as soon as there is a turn or a collision with an obstacle). Even in the breakout case, there are catastrophic translation failures for some of the studied variations although the domain gap is static and small. The image translation results look underwhelming compared to state of the art GANs used for much more complex tasks and environments (e.g., the original CycleGAN paper and follow-up works, or the ICLR'18 progressive growing of GANs paper). This might be due to a hyper-parameter tuning issue, but it is unclear why the adaptation results seem not on par with previous results although the paper is in a visually simpler domain (Atari games).

Does not address the RL generalization issues:
Although it is the main goal of the paper, the method is fundamentally side-stepping the problem as it does not improve in any way the policy or the Deep RL algorithm (they are left untouched). It is mapping the target environment to the source one, without consideration for the end task besides tuning GAN hyper-parameters. If the initial policy is very brittle (as convincingly shown in section 2), then just mapping to the source domain does not improve the generalization capabilities of the Deep RL algorithm, or even improves transfer learning: it just enables the policy to be used in other contexts that can be reduced to the training one (which is independent of the learning algorithm, RL or otherwise). So it is unclear whether the main contribution is the one claimed. The contribution seems instead an experimental observation that it might be easier to reduce related domains to the training one instead of retraining a new (specialised and brittle) policy. Existing works have actually gone further, learning jointly the image translation and task network, including for very challenging problems, e.g. in unsupervised sim-to-real visual domain adaptation (e.g., Unsupervised Pixel-Level Domain Adaptation with Generative Adversarial Networks from Bousmalis et al at CVPR'17, which is not cited here).

Experimental protocol:
The experimental conclusions are not clear and lack generality, because the optimal methods (e.g., choice of GAN, number of iterations) vary significantly depending on the task (cf. Table 3 for instance). Furthermore, the best configurations seem selected on the test set for every experiment.

Data efficiency vs actual training efficiency:
The main claim is that it is better to do image translation instead of fine-tuning or full re-training. The basis of that argument is the experimentally observed need for less frames to do the image translation (Table 2). However, it is not clear that training GANs for unsupervised image translation is actually any easier / faster. What about training instability, mode collapse, hyper-parameter tuning, and actual training time comparisons on the same hardware?



# First Recommendation

Using image translation via GANs for unsupervised domain adaptation is a popular idea, used in the context of RL for Atari games here. Although the experiments show that mapping a target visual domain to a source one can enable reusing a deep RL policy as is, the qualitative results suggest this is in fact due to a bias in the data used here and the experimental protocol does not yield general insights. Furthermore, this approach is not specific to RL and its observed generalization issues. It does not improve the learning of the policy or improve its transferability, thus having only limited new insights compared to existing approaches that jointly learn image translation and target task-specific networks in much more challenging conditions.

I believe this submission is at the start of an interesting direction, and requires further work on more challenging tasks, bigger domain gaps, and towards more joint training or actual policy transfer to go beyond this first set of encouraging but preliminary results.


# Post-rebuttal Recommendation

Thanks to the authors for their detailed reply. The clarifications around overfitting, UNIT-GAN in Section 4, and the paper claims are helpful. I  also agree that the quantitative experiments are serious. I have bumped my score by +1 as a result.

Nonetheless, the results still seem preliminary and limited in scope for the aforementioned reasons. The discussion in the comments about the learned policies and transfer are ad-hoc. A lot of the shortcomings mentioned in the review are outright dismissed (e.g., "de facto standard in RL"), downplayed (esp. generalization, which is puzzling for a transfer learning paper), or left for future work.

As there is no strong technical contribution beyond the experimental observations in the current submission, I suggest the authors try to address the GAN shortcomings both mentioned in reviews and their reply, instead of  just observing  / reporting them. As this paper's main focus is to use image translation in the proposed RL setting (with standard GAN and RL methods), I do not think it is just someone else's problem to improve the image translation part. Proposing a technical contribution there would make the paper much stronger and appealing to a broader ICLR audience.  This might also require adding a third game to ensure more generalizable experimental insights.

---

> ### Author Response · Authors · 2018-11-11
> **Authors' response to Reviewer1 - part 2**
>
> Experimental protocol:
> We agree that selecting configurations based on the test set is far from ideal, but we also note that this is the de-facto standard in video game-playing RL works, so we do not believe our work is any worse than others in the literature in this regard.
> Regarding “optimal methods (e.g., choice of GAN, number of iterations) vary significantly depending on the task” - indeed, in current GANs works, configurations (and even pre-processing in many cases) change between tasks. Given the amount of interest in GAN research, we expect this aspect to improve over time. However, we stress that in the Experiments section (Section 4) we test our approach using UNIT-GAN *only*. We test different GANs only in Section 5 as we evaluate them by comparing their results on our tasks.
>
> Data efficiency vs actual training efficiency:
> The training process of the actor-critic algorithms mainly depends on CPU where the GAN is trained using GPU therefore, you cannot compare the time on the same hardware. We agree that GANs today still suffers from many issues and are not stable enough, we also mention some of their limitation in Section 4. As noted above, we expect these aspects of GANs to improve. Despite the limitations, this method still manages to succeed in most tasks and clearly shows how transfer can be achieved by a visual mapping.
>
> More generally, we presented a novel transfer approach for model-free RL that decouples the visual transfer from the policy transfer, by using unaligned GANs. While the approach is not perfect (and we explicitly discuss many of the points raised by rev1 in the paper), the method is effective, and, to our knowledge, has not been proposed or demonstrated to work before in the context of RL. Rev1 seems to dislike the fact that we did not address the “core” problems of model-free RL directly, but rather proposed a “workaround” in the form of GAN-based mapping which is external to the RL process. In contrast, we see precisely this separation as the main idea and strength of our proposal: we let the agent re-use its learned policy by helping it map the new environment to its “previous experiences”. Furthermore, rev1 dismisses our reported quantitative results because, in their opinion, they are not reflected in the videos. However, we argue that both videos reflect the success of our approach -- the Breakout video clearly shows how the agent follows the learned policy perfectly and the Road Fighter video demonstrate how the agent applies the learned techniques from level 1 in each of the successive levels. It is possible that future iterations of the idea will improve on our method with more complex machinery, and we look forward to seeing others expand on our research.

---

> ### Author Response · Authors · 2018-11-11
> **Authors' response to Reviewer1 - part 1**
>
> Thank you for your time in reviewing our paper.
>
> Discrepancy between quantitative and qualitative results:
> While we agree that the transferred agent does not perform perfectly, we disagree that it succeeds “only due to biases in the data”. We note that (a) staying in the middle of the road is, in fact, a learned policy that took the RL agent many (a few million) game interaction frames to acquire; and (b) the agent does more than just driving at the middle of the road: it also occasionally steers to avoid obstacles, and, when hitting an obstacle such as a car, it does not immediately crash but actually succeeds to recover from the hit and keep driving in many cases. This latter behavior (recovering from hits) is also a learned skill that was discovered by the RL policy when training on the first level of the game, and which again took the agent millions of frames (and tens of thousands of crashes) to acquire. We do not consider these behaviors as “biases in the data” but rather behaviors needed to play the game. We note that, as stated in the paper, RL training requires 10M-15M iterations to achieve 5000 points in levels 2 and 4, and substantially more than that in level 3. Using our method the agent is able to apply the abilities it learned during training on level 1 (achieving scores of 5350, 5350 and 2050) after only hundred of thousands of GAN iterations and very few additional game interactions -- a significant improvement to the millions or tens of millions interaction frames needed when using RL training. Moreover, as can be seen in the videos that when using fine-tuning the agent completely fails - it crashes when hitting the sideways, even when the road is wide. Therefore, a method that overcomes this obstacle and others is considered a success. We also note that visually translating a narrow road to a wide one is, in fact, a very good strategy, provided that the position of the car on the road is correctly kept. Dealing with curve roads is more challenging, as the original agent indeed hardly observed any in its training.
>
> Regarding the deficiencies in GAN training: in Breakout, we train the model to translate images where most bricks exist and it successfully accomplishes that for similar images during testing. The problem occurs when the generator is introduced with images where many bricks are missing, images that are different from the ones it has seen during training. More generally, we indeed observed that the CycleGAN is much harder to train on seemingly easy tasks such as the breakout transfer compared to results on natural images (or even for the road-fighter game). We consistently find that such tasks are harder for GANs. The lack of diversity in our game-based datasets makes it easier for the model to overfit comparing to the datasets used in the original papers. Moreover, the fine-grained details are both much more important in the game-transfer setting *and* are easier to notice for a human observer, compared to the natural “horse to zebra” transfer where deficiencies in the background are easier to miss and easier to forgive.
>
> Does not address the RL generalization issues:
> Our paper demonstrates how the generalization problem exists in model-free deep RL algorithms due to an extreme reliance on visual details, and proposes a way to decouple the visual reliance to some extent by performing a visual mapping between related tasks. This helps to transfer the obtained non-visual knowledge across tasks. Our approach isn’t meant to solve the generalization problem, but given an overfitted model it’s a novel way to still be able to benefit from previous learning when learning a new task. We pointed out the issue of generalization and overfitting to explain the motivation for transfer approaches in this field.
>
> With regards to works such as Bousmalis et al (https://arxiv.org/abs/1612.05424) - we note that the RL setting is different than the static classification one. We did try to improve the GAN generation by adding a loss component comparing the A3C classifier results on the original and translated image, and several other approaches. These additions did not improve the results and in some cases, the results were better without them. In RL tasks, in comparison to the tasks mentioned in the Bousmalis et al, the examples are revealed only as the agent improves and goes further in the task. For this reason, we can only collect data from the early stages of the game where the optimal actions are, for example, to stay in the middle of the road and drive fast (in Road Fighter) - the exact abilities we achieved without it. The obstacles only appear in further stages, which is why this and similar domain adaptation works would not benefit to the zero-shot transfer we aim to gain. We will mention Bousmalis et al in the paper.

---

> ### Author Response · Authors · 2018-11-25
> **Re: Post-rebuttal Recommendation**
>
> Thank you for taking our comments seriously and bumping your score.
>
> We updated the paper's related work section with a discussion of Bousmalis et al and similar work, and how it differs from our learning setup and from our proposal (last paragraph of that section).
>
> We are sorry that you feel that we dismissed your concerns about the evaluation protocol, but it is indeed a de-facto standard in RL works, and we are not sure how to resolve this issue. If we misunderstood your concern, or if you have a suggestion on how to address this issue, we'd be very grateful to hear them.
>
> Regarding the generalization issue, let us attempt to clarify our argument. RL agents trained on pixels are very sensitive to the specifics of the data, and fail to generalize even to small variations. As a result, a trained agent is restricted to the data it was trained on. One strategy for attacking this problem is to adapt the RL training process to make the agent less overfitted to nuances of the training domain. This area is explored by others. We propose a different strategy, which we have not seen before, and show it is effective: rather than changing the training procedure, we use the existing agent but change how it perceives the environment by training and unaligned GAN to map between the target and source environments. The generalization across domains is performed by the unaligned GAN training.
> We show that this strategy is effective for the RL setup, both on the somewhat artificial modified-breakout setup, and on the realistic and challenging game-level transfer scenario.
>
> Finally, we indeed do not propose modified GAN or RL methods, but instead use well established techniques, which we combine in a novel way, to produce a novel result.  We do not see why this is seen as a deficiency of our work: there is already a very large number of GAN variants and RL variants, most of them do not seem to improve in any substantial way over the variants that we used, or to generalize beyond the paper that introduced them. We do not see a reason to add another small-and-quickly-forgotten variant to the literature just to seem more "technically strong" while a simple approach that re-uses existing components suffices. Our aim in this work is to propose a new method for transferring knowledge across related RL domains. We show that our proposal is vastly superior to existing approaches (fine-tuning), and also substantially more sample efficient than training from scratch, and despite the current limitations of the state-of-the-art components that we used. We believe this is sufficiently interesting without "beefing it up" with a new GAN architecture.

---

### Official Review · AnonReviewer2 · 2018-11-05
**TRANSFER LEARNING FOR RELATED REINFORCEMENT LEARNING TASKS VIA IMAGE-TO-IMAGE TRANSLATION**

**Rating:** 7
**Confidence:** 3

**Review:**

The paper seeks to generalize the reinforcement learning agents to related tasks. The authors first show the failure of conventional transfer learning techniques, then use GANs to translate the images in the target task to those in the source task. It is an interesting attempt to use the style-transferred images for generalization of RL agents. The paper is well written and easy to follow.
Pros:
1.	It is a novel attempt to use GANs to generate pictures that help RL agents transfer the policies to other related environments.
2.	It is an interesting viewpoint to use the performance of RL agent to evaluate the quality of images generated by GANS.
Cons:
1.	The pictures generated by GANs can be hardly controlled, and extra noise or unseen objects might be generated, and may fool the RL agent during training.

Other feedback:
In Figure 2, it seems the fine-tuning methods also achieve comparable results (Full-FT and Partial-FT), such as Figure 2(b) and Figure 2(c). Besides, the plot is only averaged over 3 runs, whereas the areas of standard deviation still overlap with each other. It may not be convincing enough to claim the failure of fine-tuning methods.

Minor typos:
1.	In 2.1, second paragraph: 80x80 -> $80 \times 80$
2.	In 2.1, second paragraph: chose -> choose

---

> ### Author Response · Authors · 2018-11-11
> **Authors' response to Reviewer2**
>
> Thank you for your time in reviewing our paper.
>
> * Although the results between the runs are different, in all cases the fine-tuning results don’t outperform training from scratch. We would expect a generalized model to adjust quickly to small modifications in the images since most pixels are the same as in the original games and the dynamics didn’t change. Instead, the model behaves as if the modified games are completely new tasks.
>
> * Thank you for the corrections, will change the paragraph accordingly.

---

### Official Review · AnonReviewer3 · 2018-11-05
**An interesting method to improve transfer learning between related tasks. The motivation is strong, explanations are intuitive, technical parts are solid, experiments are sufficient.**

**Rating:** 7
**Confidence:** 3

**Review:**

This paper propose an intermediate stage before transfer learning on playing new games that is with slight visual change. The intermediate stage is basically a mapping function to translate the images in the new game to old game with certain correspondence. The paper claims that the adding of intermediate stage can be much more efficient than re-train the model instead.
Then the paper compares different baselines without the mapping model. The baselines are either re-trained from scratch or (partially) initialized with trained model. The learning curves show that fine-tuning fails to transfer knowledge from the source domain to target domain. The mapping model is constructed based on unaligned GAN. And the experiments are setup and results are shown.

Pros:
+ The paper makes a very good start from analogizing human being adjusting himself between similar tasks.
+ The paper demonstrates strong motivation on improving the existing transfer learnings that are either fail or take too much time to train from scratch.
+ The paper clearly illustrate the learning curve of multiple approaches for transferring knowledge across tasks.
+ The paper proves detailed analysis why using unaligned GAN to learn the mapping model, and gives
+ I also like the experiment section. It is well written, especially the discussions section answer all my questions.

Questions:
1.	Why fine-tuning from a model that is trained from related task does not help, even decelerate the learning process? Could you explain it more?
2.	Could you please also include in figure 2 the proposed transfer learning curve with the mapping model G? I’m curious how much faster it will converge than the Full-FT. And I suppose the retrain from scratch can be extremely slow and will exceed the training epoch scope in the figure.
3.	In dataset collection, you use untrained agent to collect source domain image. Will it improve the results if you use well trained agent, or even human agent, instead?
4.	I hope, if possible, you can share the source code in the near future.

---

> ### Author Response · Authors · 2018-11-11
> **Authors' response to Reviewer3**
>
> Thank you for your time in reviewing our paper.
>
> 1. Our paper discusses tasks in which the inputs of the model in each step are raw pixels. The A3C model learned the detail and noise in the training images to the extent that small and insignificant modifications create data that is unrecognizable to the model, which prevent the model from following the policy it learned and making optimal decisions.
>
> 2. Figure 2 presents the results of the RL agents during training. In that matter, our approach is a zero-shot transfer approach in which there is no need for any additional RL training. The number of GAN iterations needed for the maximum scores and # of iterations for each task is presented in Table 1, 2.
>
> 3. Yes, if the agent is allowed to collect images from later stages of the game it needs to transfer to, results improve a little. However, we did not consider this a realistic scenario, because the untrained agent cannot reach this stages. The idea of human demonstration is an interesting one, and we expect it could work well.
>
> 4. We will share source code upon publication.

---

### Meta-Review · Area_Chair1 · 2018-12-17
**interesting result, too little conceptual contribution**

**Confidence:** 4
**Recommendation:** Reject

**Metareview:**

The paper proposes an transfer learning approach to reinforcement learning, where observations from a target domain are mapped to a source domain in which the algorithm was originally trained. Using unsupervised GAN models to learn this mapping from unaligned samples, the authors show that such a mapping allows the RL agent to successfully interact with the target domain without further training (apart from training the GAN models). The approach is empirically validated on modified versions of the Atari game breakout, as well as subsequent levels of Road Fighter, showing good performance on the transfer domain with a fraction of the samples that would be required for retraining the RL algorithm from scratch.

The reviewers and AC note the strong motivation for this work and emphasize that they find the idea interesting and novel. Reviewer 3 emphasizes the detailed analysis and results. Reviewer 2 notes the innovative idea to evaluate GANs in this application domain. Reviewer 1 identifies a key contribution in the thorough empirical analysis of the generalization issues that plaque current RL algorithms, as well as the comparison between different GAN models and finding their performance to be task-specific.

The reviewers and AC noted several potential weaknesses: The proposed training based on images collected by an untrained agent focus the data on experience that agents would see very early on in the game, and may lead to generalization issues in more advanced parts of the game. Indeed these generalization issues are one possible explanation for the discrepancies between qualitative and quantitative results noted by reviewer 1. While the quantitative results indicate good performance on the target task, the image to image translation makes substantial errors, e.g., hallucinating blocks in breakout and erasing cars in Road Fighter. To the AC, the current paper does not provide enough insight into why the translation approach works even in cases where key elements are added or removed from the scene. The paper would benefit from a revision that thoroughly analyses such cases as well as the reason why the trained RL policy is able to generalize to them.

R1 further notes that the paper does not address the RL generalization issue, but rather presents an empirical study that shows that in specific cases it is easier to translate from a target to a source domain, than to learn a policy for the target domain. The AC shares this concern, especially given the limited error analysis and conceptual insights derived from the empirical study. There are further concerns about the experimental protocol and hyper-parameter selection on the target tasks. Finally reviewer 1 questions the claim of whether data efficiency matters more than training efficiency in the proposed setting.

There is disagreement about this paper. Reviewers 2 and 3 gave high scores and positive reviews, but did not provide sufficient feedback to the concerns raised by reviewer 1, who put forward significant concerns.

The AC is particularly concerned about the experimental protocol and hyper-parameter tuning directly on the test tasks. The authors counter this point by noting that "We agree that selecting configurations based on the test set is far from ideal, but we also note that this is the de-facto standard in video game-playing RL works, so we do not believe our work is any worse than others in the literature in this regard." The AC worries about the lack of motivation to identify a strong empirical setup to arrive at the strongest possible contribution. A key concern here is that the results seem to vary substantially by task, GAN model used, etc. and substantial tuning on the target domain seems to be required. This makes it hard to draw any generalizable conclusions. This concern can be alleviated by including additional analysis, e.g., error analysis of where a proposed approach fails, or additional experiments designed to isolate the factors that contribute to a particular performance level. However, the current paper does not go to this detail of empirical exploration. Given these concerns, I recommend not accepting the paper at the current stage.